# Research on Transmission Task Static Allocation Based on Intelligence Algorithm

Xinzhe Wang [1,2,*] and Wenbin Yao [1]

1 College of Computer Science, Beijing University of Posts and Telecommunications, Beijing 100876, China
2 Radio and Television & New Media Intelligent Monitoring Key Laboratory of NRTA (Radio, Film & Television Design & Research Institute), Beijing 100045, China
* Correspondence: wangxinzhe@drft.com.cn

**Abstract:** Transmission task static allocation (TTSA) is one of the most important issues in the automatic management of radio and television stations. Different transmission tasks are allocated to the most suitable transmission equipment to achieve the overall optimal transmission effect. This study proposes a TTSA mathematical model suitable for solving multiple intelligent algorithms, with the goal of achieving the highest comprehensive evaluation value, and conducts comparative testing of multiple intelligent algorithms. An improved crossover operator is proposed to solve the problem of chromosome conflicts. The operator is applied to improved genetic algorithm (IGA) and hybrid intelligent algorithms. A discrete particle swarm optimization (DPSO) algorithm is proposed, which redefines the particle position, particle movement direction, and particle movement speed for the problem itself. A particle movement update strategy based on a probability selection model is designed to ensure the search range of the DPSO, and random perturbation is designed to improve the diversity of the population. Based on simulation, comparative experiments were conducted on the proposed intelligent algorithms and the results of three aspects were compared: the success rate, convergence speed, and accuracy of the algorithm. The DPSO has the greatest advantage in solving TTSA.

**Keywords:** task allocation; improved genetic algorithm; discrete particle swarm optimization; hybrid intelligent algorithm; probability selection model

## 1. Introduction

Task allocation is a general form of an assignment problem belonging to the combinatorial optimization problem (COP) in mathematics [1]. In the field of radio and television transmission, the problem involves the assignment of multiple transmission tasks to multiple pieces of transmission equipment at the same time. Because the equipment has different effects when performing different tasks, the comprehensive implementation effect of the static allocation of the whole task group depends on the task allocation arrangement scheme. Transmission task static allocation (TTSA) is a typical NP-hard problem.

Typical COPs include the traveling salesman problem (TSP), vehicle routing problem (VRP), and job shop scheduling problem (JSSP). There are two methods for solving a COP: an approximation algorithm and an exact algorithm. The solution time of the exact algorithm increases with the size of the problem, and an optimal solution cannot be calculated within the effective timescale for large-scale problems. It is an effective way to solve COPs by using an intelligence algorithm for the approximation algorithm, but for specific problems, the adaptive improvement of the intelligence algorithm can obtain better calculation results [2–4].

### 1.1. Related Works

Among many intelligence algorithms, the genetic algorithm (GA) has certain advantages in solving COPs because of its simple operation, fewer parameters, and high

efficiency. In recent years, there have been many studies on solving COPs using GA for specific problems [5–7].

Park et al. used the unified genetic algorithm to solve the flexible job shop scheduling (FJSSP) problem. Through the improvement of the genetic algorithm (GA) by adapting it to the specific operation of FJSSP, the algorithm performance was greatly improved [5]. Zhang et al. adopt GA based on multi-layer coding to solve the problem of parking apron support vehicle operation scheduling optimization. The authors designed a mathematical model of vehicle operation scheduling optimization adapted to GA, and improved the algorithm efficiency through multi-level coding [6]. Escamilla-Serna et al. adopted a hybrid intelligent algorithm to solve the FJSSP, established a hybrid intelligent algorithm based on GA and the random restart climbing algorithm, and realized the search in the FJSSP instance, thus obtaining the optimal solution [7]. In another study, an IGA was used to solve the problem of moving edge computing. In this approach, the population uses matrix coding, the selection operator uses a roulette wheel algorithm (the probability of being selected is proportional to the fitness), the annealing operation is completed in the selection operator replication phase, and a temperature parameter is introduced. As the temperature decreases, the number of times that the individuals of the population jump out of the local optimal average increases, and the crossover operator uses multi-row matrix hybridization and the whole row is interchanged. Finally, the mutation operator and feasibility test are completed [8]. Fan Ho-Ming et al. applied GA to solve the multi-center VRP. In the design of the two-parent genetic operator, the fixed-length random segments of two parents are used to connect, and the remaining positions are filled according to the sequence rules [9]. Belhadj et al. used GA to solve the problem of automatic repair sequencing. Because each editing operation of this problem is relatively independent, it is possible to select the random intersection point to divide the position and carry out the crossover replacement operation for the two sequences [10]. Concerning the aspect of task assembly allocation optimization, there are many cases that use GA to solve problems [11–13], all of which adopt the method of conflict processing after direct crossing. Li Mei et al. adopted GA to solve the unrelated parallel machine scheduling problem, built a mathematical model with the minimum delay time as the optimization goal, applied a genetic tabu search algorithm (GATS) to solve the problem, and compared the artificial bee colony algorithm (ABC) and genetic simulated annealing algorithm (GASA) to identify problems of different scales [14]. Tian et al. proposed a hybrid algorithm combining fuzzy simulation and GA to solve the facility location assignment (FLA) problem [15].

The above research shows that intelligence algorithms such as GA, SA, ABC, and TS can be used to solve COPs, including task allocation, FJSSP, and FLA. In the process of problem solving, intelligent algorithms are usually improved to adapt to specific problems, or use hybrid computing to improve performance.

The particle swarm optimization (PSO) algorithm has certain advantages in solving task allocation problems with its flexibility and global optimization ability [16,17]. Han et al. studied integrated production planning and scheduling with a fuzzy start time and processing time. They developed a fuzzy two-level decision technology based on the PSO algorithm and heuristic method [18]. Jamrus et al. developed a hybrid method combining the PSO algorithm and genetic operator to solve the fuzzy shop scheduling problem with an uncertain processing time [19]. For the scheduling problem of complex products in multi-shop production, Qiao et al. proposed the multi-level process network diagram construction method and built a comprehensive mathematical model of multi-shop production scheduling. An improved PSO algorithm was proposed to solve this problem. By constructing the network subgraph, the invalid search path of the algorithm is avoided, and the efficiency of the algorithm is improved. For scheduling problems with product time constraints, a path search rescheduling strategy was proposed to ensure that the algorithm can obtain an effective search path. Through the analysis of complex product scheduling in the multi-shop environment, the effectiveness and practicability of the above methods were verified [20]. Zhang Yan-Me et al. used PSO to solve the problem of arrangement

optimization in the field of software testing. By mapping the class test sequence to a one-dimensional space to form a particle position code, the speed and displacement of each particle were calculated according to the fitness function, and then the PSO algorithm was used to select the optimal position and fitness of the particles to obtain the optimal particle. Finally, according to the mapping relationship, the optimal particle code was mapped to the optimal class test sequence [21]. Zhou Ya-lan et al. proposed the use of the distribution estimation discrete particle swarm optimization (DPSO) algorithm to solve the permutation problem, and combined the PSO and the distribution estimation algorithm (EDA) to solve the permutation optimization problem, breaking through the speed-displacement update mode of the standard PSO algorithm. Based on the PSO idea, the specific update operation was processed by the distribution estimation algorithm, so as to calculate the optimal solution of the permutation COP [22]. Ma Ding et al. used the DPSO algorithm to solve the multi-objective service path construction problem, and proposed a new particle location initialization and update strategy, which effectively optimized the quality of the service path compared with existing algorithms [23]. Li He et al. proposed the combination of DPSO and non-dominated sorting GA based on elite strategy (multi-objective optimization), introduced the idea of PSO, and completed the update and iteration based on genetic operations at the bottom [24]. For the rectangular optimal layout problem (a permutation coding COP), a hybrid PSO algorithm was proposed. The concept of commutator and commutation order was introduced to solve the problem that the particle update is difficult to describe when the standard PSO solves the COP, and to improve the convergence speed of the processing algorithm in application problems [25]. In addition, research using cluster search algorithms has included solving the disassembly line balancing problem (DLBP) based on the discrete whale swarm algorithm [26], and solving human–robot collaborative disassembly (HRCD) based on the discrete bees algorithm [27,28].

The above research shows that PSO can be used to solve discrete problems through certain adaptive improvements, but there is no universal PSO solution for different specific application problems. To address the problem that particle swarm operations are difficult to describe, this paper first proposes a hybrid intelligent algorithm using improved genetic crossover operators instead of particle movement operations; second, a probability selection model for alternative vector computation and the definition of the movement direction and target with the largest D-value are proposed, both of which are not mentioned in the abovementioned literature.

### 1.2. Contributions

When using an intelligence algorithm to solve specific COPs, the applicability of the algorithm is a difficult problem, and the calculation results from different intelligent algorithms vary considerably. To solve the problem of TTSA, the following problems need to be solved:

1.  Establishing a mathematical model suitable for intelligent algorithm calculation;
2.  Selecting an appropriate intelligent algorithm based on the characteristics of the problem;
3.  Making targeted improvements to the algorithm to improve its success rate and efficiency.

Based on the research regarding the application of GA and PSO in COPs, this paper studies the TTSA and identifies an intelligence algorithm that can obtain accurate results within a certain range through the comparison of multiple algorithms.

The main contributions of this paper are as follows:

1.  According to the characteristics of TTSA, a mathematical model suitable for the application of intelligent algorithms is built, a matrix of evaluation values of the effect is constructed, and the input data, output data, and fitness function of the algorithm are defined;

2.    Based on GA, an improved crossover operator based on cyclic substitution grouping is proposed to avoid the loss of excellent chromosome genes due to conflict processing and to improve the execution efficiency of IGA;
3.    Based on GA and PSO, two hybrid intelligent algorithms are proposed to provide more intelligent algorithm ideas for solving TTSA;
4.    A DPSO is proposed. Based on TTSA, the position and direction of the particles are described again, the probability selection model is used to update the particle position, and the random disturbance strategy is added to improve the particle inertia retention.

In Section 2, we build a mathematical model of TTSA suitable for intelligent algorithm processing. In Section 3, based on GA and PSO, the four intelligent algorithms proposed are described. In Section 4, the experimental results are presented, and the results are analyzed and discussed. In Section 5, we present the conclusion and make recommendations for future work.

## 2. Preliminaries

### 2.1. Model of TTSA

The problem of TTSA exists in the automatic control system of radio and television transmission stations. The data acquisition of the transmission equipment is completed by the computer. According to the analysis of the transmission effect data, the appropriate transmission tasks are assigned to the transmission equipment with the best effect, and then the automatic transmission is completed by the computer control [29–32]. The TTSA problem is to solve the optimal allocation scheme between the task and the equipment performing the task. The problem framework is shown in Figure 1.

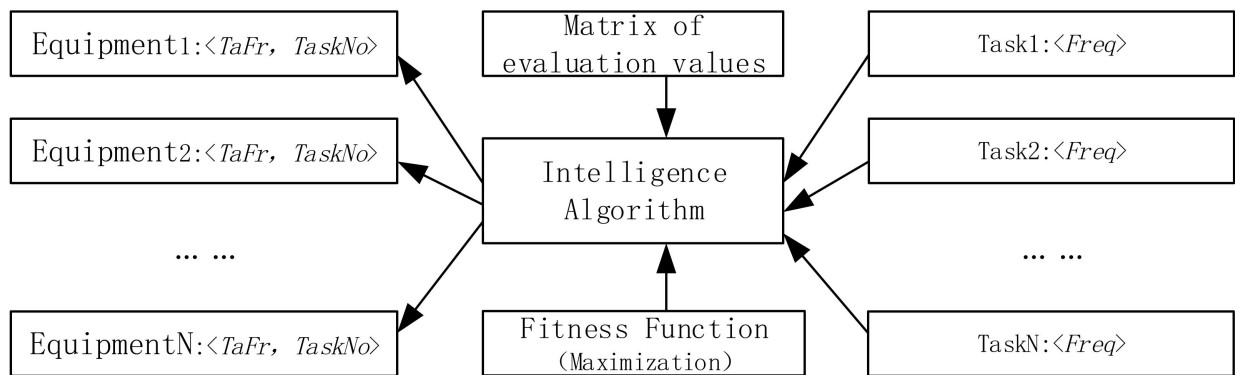

**Figure 1.** Framework model for TTSA.

In Figure 1, the task queue is composed of *N* tasks, each with different frequencies. The matrix of evaluation value is the basis for the evaluation of the transmission effect. It stores the evaluation value of the transmission effect obtained when each piece of equipment works in different transmission frequency ranges. The initial data are set according to the data collected by polling the transmission effect during the test run of the transmission station. Combined with the idea of supervised learning, the matrix of evaluation value modifies the matrix according to the collected data after the daily transmission task to ensure the practicability of the algorithm. Each transmission equipment item can only complete one transmission task at a time. In order to ensure the integrity of the algorithm, this paper sets the number of transmission tasks to be consistent with the number of transmission equipment items.

#### 2.1.1. Input Data

According to the functional characteristics of the transmission equipment, the transmission frequency range of the equipment is described as $[Fr_{min}, Fr_{max}]$: the operating frequency of each piece of equipment is divided into *m* frequency bands. According to the

actual situation, it can be assumed that the transmitting effect of the equipment performing tasks on the frequencies within the frequency band range is consistent. The mathematical model of frequency band division is as follows:

$$FreqR = \{Fr_1, Fr_2, \ldots, Fr_i, \ldots Fr_{m-1}\}, \quad Fr_i \in [Fr_{min}, Fr_{max}] \tag{1}$$

where $m$ is the number of frequency bands and $Fr_i$ is the frequency boundary value between the $(i-1)$th and $i$th frequency bands.

For the medium- and short-wave transmitter performing the transmission task, the transmission task command parameters include the working frequency, power level, and working type. The values of the working type and other parameters are independent of the optimal allocation of the task. In order to simplify the mathematical model, different transmission tasks are only distinguished according to the transmission frequency. At the same time, the transmission task set consists of n frequencies, and the transmission task can be described as:

$$Task = \{Freq_1, Freq_2, \ldots, Freq_i, \ldots Freq_n\} \ Freq_i \in [Fr_{min}, Fr_{max}] \tag{2}$$

where $n$ is the number of transmission tasks, and $i$ is the $i$th task expressed by the frequency value.

The working frequency of the task belongs to the working frequency band divided by the model, and the calculation formula is as follows:

$$TaFr_i = \begin{cases} 0, & Freq_i \in [Fr_{min}, Fr_1] \\ k, & Freq_i \in (Fr_k, Fr_{k+1}] \\ m, & Freq_i \in (Fr_m, Fr_{max}] \end{cases} \tag{3}$$

Through the calculation of Formula (3), the set $TaFR$ of the working frequency band of the task is obtained, which can be described as follows:

$$TaFR = \{TaFr_1, TaFr_2, \ldots, TaFr_i, \ldots TaFr_n\}, TaFr_i = 0, 1, \ldots, m \tag{4}$$

where $n$ is the number of transmission tasks, and $i$ is the segment code obtained by dividing the $i$th task by frequency segment.

The implementation effect of the evaluation task on the equipment is obtained through data quantitative calculation. The rated power of the transmitting equipment is recorded as $R_{max}$. When the equipment is set to transmit at full power, the monitoring data collected by the receiving equipment are: forward power, recorded as $R$; reverse power, recorded as $r$. The transmission effect evaluation in this paper is mainly based on the transmission standing wave ratio ($SWR$), and the transmission effect calculation formula is as follows:

$$SWR = \frac{\sqrt{R} + \sqrt{r}}{\sqrt{R} - \sqrt{r}} \tag{5}$$

According to the characteristics of the transmitting equipment, the minimum value of the transmitting $SWR$ is not less than 1.0, and the closer it is to 1.0, the better the effect. The formula is as follows:

$$Va = \frac{1}{SWR} \tag{6}$$

To sum up, *ValueMatrix* can be obtained through the transmission test run of each piece of transmission equipment in each frequency band, which can be described as:

$$ValueMatrix = \begin{bmatrix} Va_{11} & Va_{12} & \cdots & Va_{1i} & \cdots & Va_{1m} \\ Va_{21} & Va_{22} & \cdots & Va_{2i} & \cdots & Va_{2m} \\ \cdots & \cdots & \cdots & \cdots & \cdots & \cdots \\ Va_{i1} & Va_{i2} & \cdots & Va_{ij} & \cdots & Va_{im} \\ \cdots & \cdots & \cdots & \cdots & \cdots & \cdots \\ Va_{n1} & Va_{n2} & \cdots & Va_{nj} & \cdots & Va_{nm} \end{bmatrix} \tag{7}$$

where *n* is the number of transmission tasks. In this paper, the number of transmission equipment items *n* is the number of frequency bands divided, and *m* is the evaluation value of the transmission effect of the *i*th transmission equipment working in the *j*th frequency band when considering the optimal allocation of tasks in the same period.

### 2.1.2. Output Data

In practice, the operating frequency band of the transmission task that the transmission equipment can complete has some constraints. This paper aims to solve the problem of task optimization allocation. In the model, it is assumed that each transmission equipment item can complete the transmission task of any one of the transmission frequency bands. The transmission equipment can be described as $Tr_j$, including the following data:

$$Tr_i = \langle TaFr, TaskNo \rangle, i = 0, 1, \ldots, n, \ TaFr = 0, 1, \ldots, m, \ TaskNo = 0, 1, \ldots, n \tag{8}$$

The $Tr_i.Tafr$ is the working frequency band of the *i*th equipment task, and represents the transmission task code undertaken by the current transmission equipment. The sequence of the results expressed by the complete output mathematical model of the algorithm is as follows:

$$Result = \left\{ Tr_1.TaskNo, Tr_2.TaskNo, \ldots, Tr_j.TaskNo, \ldots Tr_n.TaskNo \right\} \tag{9}$$

where *j* is the task sequence code of the transmission task assigned to the *j*th transmission equipment item.

### 2.1.3. Fitness Function

The objective function of the task optimization allocation of the above output results is the maximum value of the transmission effect evaluation, which can be described as:

$$Max: \ Value(Result) = \frac{\sum_{i=1}^{n} Va_{i \ Tr_i.TaFr}}{n} \tag{10}$$

According to the *Result* of the task allocation in the sequence, the evaluation value of each equipment allocation task is calculated and averaged to obtain the evaluation value of the entire result sequence. In the formula, *i* is the equipment sequence number, $Tr_i.Tafr$ is the operating frequency band of the *i*th equipment transmission task, and $Va_{i \ Tr_i.TaFr}$ represents the comprehensive evaluation value of the *i*th equipment operating in the frequency band $Tr_i.TaFr$ in the comprehensive evaluation *ValueMatrix* of the transmission effect.

According to the above model framework and data model description, the data structure parameters are as outlined in Table 1.

### 2.2. GA

The GA can be well applied to most COPs. The GA takes all individuals in the population as the object, and uses randomization to search an encoded parameter space efficiently. Among them, selection, crossover, and mutation constitute the basic genetic operations of GA. The core content of GA is composed of five elements: parameter coding, initial population setting, fitness function design, genetic operation design, and control

parameter setting. As a global optimization search algorithm, GA is simple, universal, robust, and suitable for solving COPs.

**Table 1.** TTSA parameter table.

| Parameter | Explanation |
|---|---|
| $Freq_i$ | Transmission frequency value of the task |
| $Task$ | Input data and transmission task sequence |
| $Va_{ij}$ | Evaluation value of the transmission effect of the $i$th transmission equipment operating in the $j$th frequency band |
| $ValueMatrix$ | Transmission effect evaluation value matrix |
| $Tr_i$ | Description of the $i$th transmitting equipment, including $\langle TaFr, TaskNo \rangle$ |
| $Result$ | Output results, consisting $n$ of $Tr_i.TaskNo$ ordered sequences |

In the design of the mathematical model, some of the five elements of GA were set, including the use of a task allocation result queue to solve the problem of chromosome coding parameters, and the use of an optimal allocation objective function to solve the design of fitness function. According to the summary of relevant research, the GA has been used to obtain a large number of research results in dealing with COPs, including the selection operator based on strategies such as championship, roulette, and elite reservation, and the crossover operator based on technologies such as multi-agent GA, multi-point crossover, random crossover, and conflict processing, but it is also used to solve TTSA. The adaptive improvement of genetic operation and control parameters is also needed.

*2.3. PSO*

PSO is a group search optimization algorithm. The motion of each particle is determined by the value of the fitness function, and the "direction" and "displacement" of its motion are determined by the "speed" of each particle. Then, the particles iterate in the solution space according to the direction of the current optimal particle.

In PSO, $x$ represents the position of the particles, $v$ represents the motion speed of the particles, and $p$ represents the optimal position of the particles searched. PSO initializes a group of random particles and finds the optimal solution through iteration. In each iteration, the particle updates its position by tracking two optimal values. One optimal value is the optimal solution that the particle can find. This solution is called particle best; the other optimal value is the optimal solution found by the whole population at present, which is called the global best. Suppose that a population composed of $K$ particles is searched in the D-dimensional solution space, where the position of the $i$th particle is expressed as a D-dimensional vector:

$$X_i = (x_{i1}, x_{i2}, \ldots, x_{iD}) \ , i = 1, 2, \ldots, K \tag{11}$$

The motion speed of the $i$th particle is also a vector of the D-dimension:

$$V_i = (v_{i1}, v_{i2}, \ldots, v_{iD}) \ , i = 1, 2, \ldots, K \tag{12}$$

The optimal position searched by the $i$th particle, namely, the particle best, is expressed as:

$$Pbest_i = (p_{i1}, p_{i2}, \ldots, p_{iD}) \ , i = 1, 2, \ldots, K \tag{13}$$

The optimal position searched by the whole population, namely, the global best, is expressed as:

$$Gbest = (g_1, g_2, \ldots, g_D) \tag{14}$$

The update formula of speed and position is as follows:

$$v_{id} = w * v_{id} + c_1 r_1 (p_{id} - x_{id}) + c_2 r_2 (g_d - x_{id}), \ d = 1, 2, \ldots, D \tag{15}$$

$$x_{id} = x_{id} + v_{id}, \, d = 1, 2, \ldots, D \tag{16}$$

where $c_1$ and $c_2$ are the acceleration constant, $r_1$ and $r_2$ are uniform random numbers, and $w$ is the inertia constant.

### 3. Methodologies

#### 3.1. IGA

In order to adapt to TTSA, the genetic selection operator, genetic crossover operator, and genetic mutation operator are improved, as follows:

1. The selection operator (SO) adopts the elitist retention strategy and increases the parameter of selection factor. When selecting the population, it retains a certain proportion of excellent parents, and enters the selection range together with the offspring, so as to achieve the optimal individual survival rate in the process of genetic evolution. In order to ensure the diversity of the population, the algorithm replaces the duplicate individuals in the new population during the execution of the selection operator in each iteration.

2. The improved crossover operator (ICO) adopts the traditional genetic crossover processing according to the arrangement sequence that each individual is assigned to the equipment based on the task. Usually, the crossover point is determined, the sequence before the crossover point is reserved, and the sequence after the intersection point of two individuals is exchanged to generate new individuals. However, for two permutation sequences, there is a high probability of conflict after cross-processing, which is reflected in the issue of TTSA; that is, one task is assigned to two equipment items or two tasks are assigned to one equipment item. This paper proposes a method of the cyclic exchange of packets. The two sequences are divided into multiple packets according to the cyclic exchange calculation between two parents using a tracking calculation, and then one or more packets whose size is close to half of the length of the individual sequence are selected as the reservation or exchange reason, thus avoiding the occurrence of node gene conflict after the crossover operation and achieving the purpose of preserving all excellent genes. The specific operation is shown in the example provided in Figure 2.

3. The improved mutation operator (IMO): In order to ensure the diversity of the population and avoid falling into the local optimal trap, the mutation operator uses the random selection of two positions in the sequence to exchange the transmission tasks performed by the two positions of the transmission equipment to generate new mutation individuals. The specific operation is shown in the example provided in Figure 3.

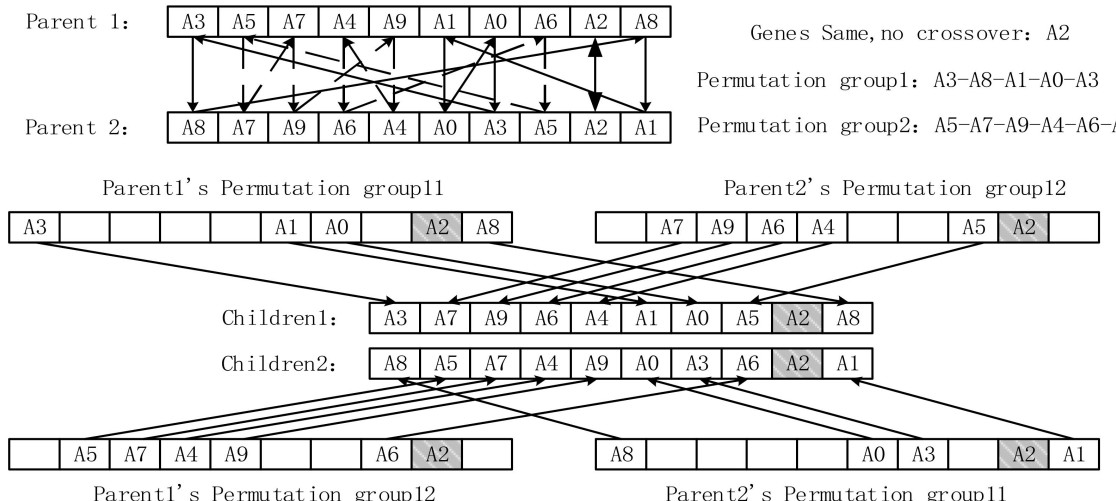

**Figure 2.** Process of ICO.

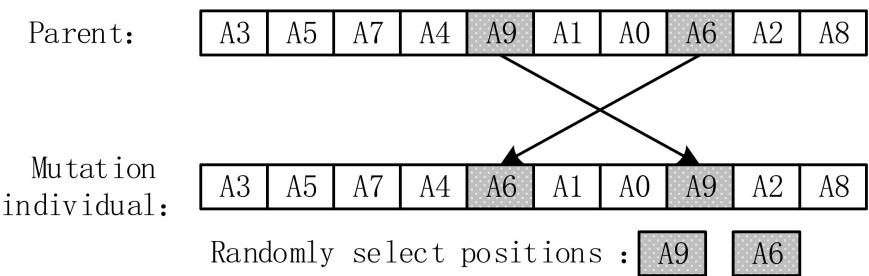

**Figure 3.** Process of IMO.

*3.2. PSO-GA Hybrid Algorithm*

The idea of PSO is introduced to improve the selection operator, crossover operator, and mutation operator of IGA.

1.  Selection operator: PSO is adopted to formulate a genetic selection strategy and establish the PSO population. The update operation of each particle selects global best, particle best, and particle position for the crossover operator.
2.  Crossover operator: The ICO operator above is used for the crossover operation, and the idea of multi-agent inheritance is introduced. Three gene sequences are selected based on the selection operator, and the particle position is crossed with the global extreme and individual extreme, respectively, to generate four generations. Then, the four children are divided into two groups. The two children with a global best gene and the two children with a particle best gene cross over each other to generate eight children. The children with the largest evaluation value are selected from the eight children as the result of the crossover calculation to replace the particle position and complete the iteration. The specific operation is shown in Figure 4.
3.  Mutation operator: In order to ensure the diversity of the population and avoid falling into the local optimal trap, the IMO above performs a mutation operation on the particle position. The mutation factor is set, and the particle position of the population is calculated according to the factor proportion according to the gene mutation operation.

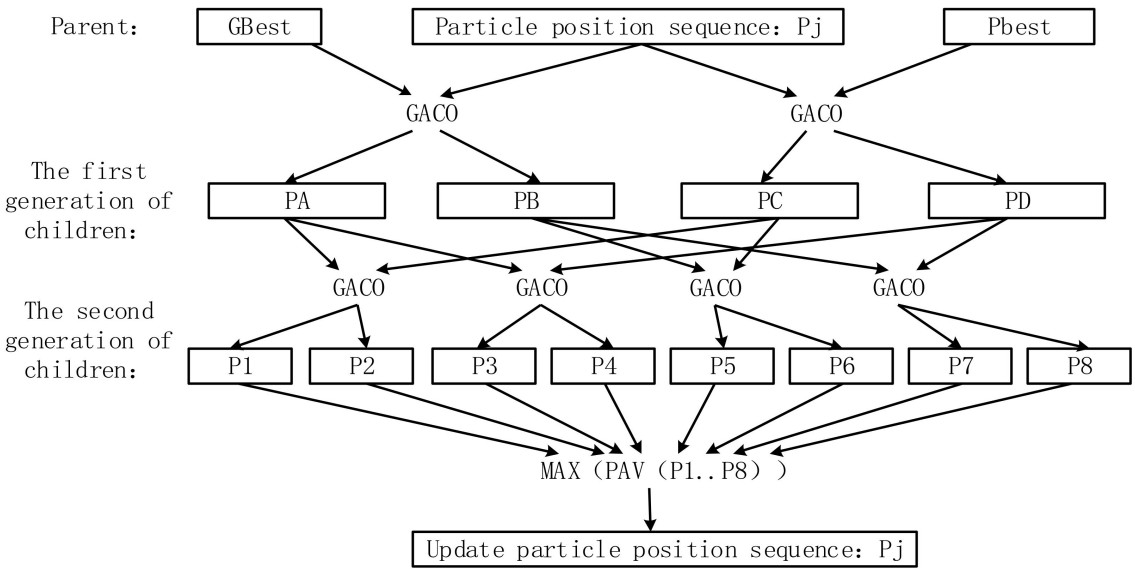

**Figure 4.** Selection operator and particle update operation.

The PSO-GA only sets one parameter of mutation factor (MF), which is easy to operate and easy to implement.

*3.3. GA-SPO Hybrid Algorithm*

3.3.1. Probability Selection Model and Random Disturbance

In the basic PSO, the speed update is composed of three parts: the inertia maintenance part, the particle best direction update part, and the global best direction update part. In Formula (15), $w, c_1 r_1, c_2 r_2$ represent the influence of the three parts of the update on the final speed update. For continuous problems, the final particle speed can be obtained by vector calculation and parameter comprehensive reference. However, the TTSA has discrete characteristics. Inertia retention, particle best, and global best represent different task allocation schemes. The schemes are discrete from each other, and there is no correlation between them. Therefore, the vector superposition method cannot be used for calculation. The probability model is used to select one of the three parts as the final selected particle motion direction and target; that is, the algorithm parameters $w, c_1 r_1, c_2 r_2$ are set according to the probability model, which respectively represent the probability value of selecting the inertia retention, particle best, and global best, and the final direction and target are selected according to the set probability value proportion. The $w, c_1 r_1, c_2 r_2$ total probability is 100%, and the $w$ probability is inertia retention operation. The probability $c_1 r_1$ is updated according to the particle best direction and target, and the $c_2 r_2$ probability is updated according to the global best direction and target.

According to the definition of the PSO, inertia is maintained as the particles continue to move forward along the direction and displacement of the last iteration. The selection of the last iteration has been replaced and updated according to the rules. Inertia maintenance has no practical operational significance.

In order to avoid the particle position update falling into the local optimal solution trap, a random perturbation operation is proposed: when the particle motion direction and target calculated in this iteration are completely consistent with the motion direction and target calculated last time, the algorithm randomly selects the task nodes to exchange, thus increasing the diversity of the particle position update and avoiding the algorithm falling into the local optimal solution trap.

3.3.2. GA-PSO Operation

Based on PSO, because the speed and position update operation of the defined particles is complex, the ICO is used to replace the update operation of PSO, and the parameter control algorithm of GA-PSO is used for execution. The operational improvements are as follows:

1.  Definition of particle position: According to the characteristics of TTSA solution space of the transmission task, the particle swarm coding is represented by the preceding text; that is, the "position" of the particle is represented by an orderly task sequence number. When the algorithm is started, a random sequence is used to initialize the particles. The goal of the algorithm is to continuously update the particle's position, so that the task allocation sequence represented by the position information can obtain the maximum comprehensive effect evaluation value through Formula (10)—that is, continuously optimize the Result sequence, and obtain the optimization result of the particle swarm optimization, which is recorded as:

$$X_i = (x_{i1}, x_{i2}, \ldots, x_{ij}, \ldots, x_{in}), i = 1, 2, \ldots, K, x_{ij} = Tr_j.TaskNo \tag{17}$$

The definition of particle best and global best is the same as the definition of particle position, which is recorded as:

$$Pbest_i = (p_{i1}, p_{i2}, \ldots, p_{ij}, \ldots, p_{in}), i = 1, 2, \ldots, K \quad p_{ij} = Tr_j.TaskNo \tag{18}$$

$$Gbest = (g_1, g_2, \ldots, g_i, \ldots, g_n), \quad g_i = Tr_i.TaskNo \tag{19}$$

2.  Particle update operation based on probability selection and ICO: When the particle branch direction is calculated according to the parameters, and inertia operation is

performed according to the parameters, some particles move to global best, and some particles move to particle best. In order to avoid the problem whereby the motion direction and speed cannot be defined by a vector, the ICO is used to replace the particle motion operation for the move toward global best and particle best. One of the two generations is selected with a better evaluation value to replace the particle position to realize particle iteration. This achieves the goal of simplifying the PSO. The particles performing inertia operations adopt MO to increase the diversity of the particles.

3.  Algorithm termination: When the defined number of iterations is reached, the algorithm terminates, and the global best is the output of the algorithm's optimal solution, ending the algorithm calculation.

### 3.4. DSPO

#### 3.4.1. Particle Motion Direction and Target Definition

The PSO is usually used to solve the optimization problem of continuous space. The problem of the optimal allocation of transmission tasks mainly involves solving the mapping problem between the tasks and equipment, which is obviously a discrete allocation problem. The solution space of this problem is equivalent to the total arrangement COP of allocating $n$ tasks to $n$ equipment items, and the number of all solutions in the space is $P_n^n$.

According to the previous description of the PSO, the PSO needs to be improved to adapt to the characteristics of the problem. The particle motion direction and target are redefined.

1.  Definition of particle motion direction: According to the description of the PSO, the particle motion speed is calculated by the position vector, which contains the direction and displacement information. In TTSA, due to the discrete characteristics, the decomposition speed is divided into two items: direction and displacement. Each node of the task sequence is defined as the particle motion direction. The particle motion direction suitable for the problem in this paper is $VD_i$, defined as:

$$VD_i = \left(vd_{i1}, vd_{i2}, \ldots, vd_{ij}, \ldots, vd_{in}\right), \ i = 1, 2, \ldots, K, \ vd_{ij} = 0, 1 \tag{20}$$

Only one $vd_{ij}$ of the $VD_i$ sequence values represented by each is 1, and the other is 0. The position of 1 represents the task of replacing the node when the particle updates its position, that is, the particle's motion direction.

2.  Definition of particle motion target: According to the discrete characteristics of TTSA, the velocity displacement is defined as the number of transmission tasks to be replaced by the node representing the particle motion direction. Referring to the PSO Formula (12), the particle motion target suitable for the problem in this paper is defined as:

$$VT_i = \left(vt_{i1}, vt_{i2}, \ldots, vt_{ij}, \ldots, vt_{in}\right), \ i = 1, 2, \ldots, K, \ vt_{ij} = 0, TaskNo \tag{21}$$

Only one $vt_{ij}$ value of each represented $VT_i$ sequence is taken as *TaskNo*, and the others are taken as 0. The value taken as *TaskNo* indicates that the particle replaces the node with the representative task when updating the position, that is, the replacement target of the particle position code in the particle motion direction, which is defined as the particle motion target.

#### 3.4.2. Particle Position Update Calculation

According to the definition of particle motion direction and particle motion target, a new update strategy of DPSO is proposed. The particle velocity update operation is to calculate the particle motion direction and the particle motion target. The emission effect

evaluation value of each task of the particles is calculated according to the particle position, which is recorded as $XV_i$:

$$XV_i = (xv_{i1}, xv_{i2}, \ldots, xv_{ij}, \ldots, xv_{in}), \quad i = 1, 2, \ldots, K, \quad xv_{ij} = Va_{j\,x_{ij}} \tag{22}$$

where $x_{ij}$ is the currently assigned task code of the $j$th equipment represented by the $i$th particle in the particle position definition, and $Va_{j\,x_{ij}}$ represents the transmission effect evaluation value in the effect evaluation matrix when the $j$th equipment executes the $j$th task. Similarly, the transmission effect evaluation value sequence of particle best and global best is recorded as $PbestV_i$, $GbestV$:

$$PbestV_i = (pv_{i1}, pv_{i2}, \ldots, pv_{ij}, \ldots, pv_{in}), \ i = 1, 2, \ldots, K \quad pv_{ij} = Va_{jp_{ij}} \tag{23}$$

$$GbestV = (gv_1, gv_2, \ldots, gv_i, \ldots, gv_n), \quad gv_i = Va_{j\,g_i} \tag{24}$$

where $p_{ij}$ is the task code of the $j$th equipment indicated in the definition of the particle best position of the $i$th particle, and $Va_{jp_{ij}}$ represents the transmission effect evaluation value in the effect evaluation matrix when the $j$th equipment performs the $p_{ij}$ task. Similarly, it is the $g_i$ task code of the $j$th equipment indicated in the global best definition, and $Va_{j\,g_i}$ represents the transmission effect evaluation value in the effect evaluation matrix when the $j$th equipment executes the $g_i$ task.

The DPSO calculates the D-value between the current particle and the particle best according to the D-value between the two data series of Formulas (22) and (23), and calculates the D-value between the current particle and the global best according to the D-value between the two data series of Formulas (22) and (24). The maximum D-value is selected, that is, the maximum D-value between the evaluation values, as the motion direction option of particles. At the same time, the position of the particle best and the global best of the maximum D-value position is recorded, that is, the task code of the corresponding position in Formulas (20) and (21).

The selection of particle best and global best selects the iteratively updated particle motion direction according to the probability model, and the task code is the particle motion target.

Taking particle best as an example, through the comparison of $XV_i$ and $PbestV_i$ calculations, the maximum value of $Va_{jp_{ij}} - Va_{j\,x_{ij}}$ at the time $j = L$, and then $L$ is determined as the motion direction calculated according to the particle best, that is, only $VD_i = (0, 0, \ldots, 1, \ldots, 0)$, the $L$th value is 1, and the other value is 0. The motion target is $VT_i = (0, 0, \ldots, Tr_L.TaskNo, \ldots, 0)$, where $Tr_L.TaskNo$ is the task code with the value in the $L$-th position in Formula (18).

A summary of the DPSO parameters proposed to solve TTSA are presented in Table 2.

**Table 2.** DPSO parameter table.

| Parameter | Explanation |
|---|---|
| $XV_i$ | Particle evaluation value sequence: the evaluation value sequence taken from the task allocation sequence represented by the $i$th particle |
| $PbestV_i$ | Particle best evaluation value sequence: the evaluation value sequence obtained from the task allocation sequence recorded by the $i$th particle best |
| $GbestV$ | Global best evaluation value sequence: the evaluation value sequence obtained from the task allocation sequence of the global best record |
| $w$ | In the probability selection model, the inertia maintains part of the probability value. According to the DPSO, this part is the random disturbance probability value |
| $c_1 r_1$ | In the probability selection model, particle best partial probability value |
| $c_2 r_2$ | In the probability selection model, global best partial probability value |

## 4. Experiments

### 4.1. Simulation Experiment Environment

The experimental program writing tool is Visual Studio 2012, and the language is C++ with MFC architecture. The hardware environment is a Microsoft Surface X1 portable computer, the CPU is Intel Core i7 3.60 GHz, the memory is 16 GB, and the operating system is Win10.

In order to test the effect of the algorithm, the experiment adopts the full task allocation mode with an equal number of equipment items and tasks. The task data are randomly generated, and the frequency band is divided into eight bands. The evaluation value matrix uses randomly generated fixed values. According to the actual data acquisition simulation of the *SWR*, the evaluation value is randomly generated in the range of 0.5–1.0. In order to compare the efficiency and results between the different algorithms and different configuration parameters, the same task sequence and evaluation value matrix are used for the input, and the average value of multiple tests is used for comparison.

### 4.2. Comparison Algorithms

The research on TTSA is relatively sparse, and GA and the various intelligent algorithms proposed are compared. The specific operation of GA refers to the specific methods in the reference literature and the reprograms based on the application problems in this paper. The specific genetic operation is described in Table 3. The other four intelligent algorithms are proposed in this paper.

**Table 3.** Compare intelligent algorithm list.

| Algorithm | Explanation | Parameter Representation | Parameter Description |
|---|---|---|---|
| GA | Selection: roulette strategy, Crossover: direct crossover and then conflict processing strategy, Mutation: random mutation and then conflict processing strategy [8,11–13] | GA (CF, MF) | New individuals are generated in the new population according to the ratio of crossover factor (CF) and mutation factor (MF) |
| IGA | Selection: the elitist retention strategy, Crossover: replace the group crossover with the crossover cycle, Mutation: use the random exchange | IGA (SF, CF, MF) | In the new population, excellent individuals of the parent generation are retained according to the selection factor (SF), and new individuals are generated according to the ratio of crossover factor (CF) and mutation factor (MF) |
| PSO-GA | Establish a selection mechanism based on PSO, establish a multi-agent genetic mechanism based on particle itself, particle best, and global best, and use ICO and MO of IGA | PSO-GA (MF) | After the particle update operation, some particles are changed according to the proportion of the mutation factor (MF) |
| GA-PSO | Initialize the population based on PSO, according to the probability selection model, replace the particle position update with ICO, and perform the MO when maintaining inertia | GA-PSO (IRF, PBF, GBF) | According to the probability model, determine the proportion of particles in the population updated according to three parameters: inertia retention factor (IRF), particle best factor (PBF), global best factor (GBF) |
| DPSO | Initialize the population based on PSO, according to the probability selection model, and complete the position update operation. When the selected direction and target remain unchanged, add random disturbance | DPSO (IRF, PBF, GBF) | According to the probability model, determine the proportion of particles in the population updated according to three parameters: inertia retention factor (IRF), particle best factor (PBF), global best factor (GBF) |

### 4.3. Algorithm Parameter Test

Before algorithm comparison and verification, the optimal parameter selection test of each algorithm is carried out. The experimental population size of the algorithm is calculated by 100, and the population size is kept unchanged during the iteration process. The upper limit of the number of iterations of the intelligent algorithm is set according to the following formula:

$$IterationNumber = \sum_{i=1}^{n} i^2 \tag{25}$$

The value of *n* is the number of equipment items and tasks.

The algorithm parameter test is carried out using the comparison method of accurate calculation (enumeration algorithm). The parameters are selected based on the algorithm success rate (the proportion of the global optimal solution obtained by the algorithm through multiple calculations) and the algorithm convergence (after multiple calculations, the total number of iterations of the algorithm to obtain the global optimal solution is accumulated according to the upper limit of the number of iterations if the number of iterations reaches the upper limit, but still does not obtain the global optimal solution). The efficiency of intelligent algorithms depends on the calculation time of the fitness function, and the number of calculations of the fitness function is consistent in each iteration. Therefore, this paper uses the number of iterations to measure the efficiency of each intelligent algorithm. The test of the optimal parameters of each algorithm and the representative (all successful, fast convergence) parameter data are shown in Tables 4 and 5, which list the best parameter combinations of the intelligent algorithm parameter groups.

**Table 4.** Multi-algorithm parameter experiment success rate data table.

| Algorithm and Parameter | Number of Equipment Items and Tasks | | | | | Average |
|:---:|:---:|:---:|:---:|:---:|:---:|:---:|
| | 8 | 9 | 10 | 11 | 12 | |
| GA (0.6,0.4) | 100% | 99% | 100% | 100% | 88% | 97.4% |
| IGA (0.8,0.1,0.1) | 100% | 100% | 100% | 100% | 100% | 100% |
| IGA (0.7,0.2,0.1) | 100% | 100% | 100% | 100% | 100% | 100% |
| IGA (0.6,0.2,0.2) | 100% | 100% | 100% | 100% | 100% | 100% |
| IGA (0.5,0.3,0.2) | 100% | 100% | 100% | 100% | 100% | 100% |
| IGA (0.4,0.3,0.3) | 100% | 100% | 100% | 100% | 100% | 100% |
| IGA (0.3,0.3,0.4) | 100% | 100% | 100% | 100% | 100% | 100% |
| PSO-GA (0.6) | 100% | 99% | 97% | 94% | 95% | 97% |
| GA-PSO (0.3,0.1,0.6) | 100% | 100% | 100% | 100% | 100% | 100% |
| GA-PSO (0.1,0.2,0.7) | 100% | 99% | 100% | 100% | 99% | 99.6% |
| GA-PSO (0.6,0.2,0.2) | 100% | 100% | 100% | 99% | 99% | 99.6% |
| DPSO (0.2,0.3,0.5) | 100% | 100% | 100% | 100% | 100% | 100% |
| DPSO (0.3,0.2,0.5) | 100% | 100% | 100% | 100% | 100% | 100% |
| DPSO (0.1,0.2,0.7) | 100% | 100% | 100% | 100% | 100% | 100% |
| DPSO (0.5,0.1,0.4) | 100% | 100% | 100% | 100% | 100% | 100% |
| DPSO (0.4,0.1,0.5) | 100% | 100% | 100% | 100% | 100% | 100% |
| DPSO (0.3,0.1,0.6) | 100% | 100% | 100% | 100% | 100% | 100% |

**Table 5.** Multi-algorithm parameter experiment number of iteration data table.

| Algorithm and Parameter | Number of Equipment Items and Tasks | | | | | Total |
|---|---|---|---|---|---|---|
| | 8 | 9 | 10 | 11 | 12 | |
| GA (0.6,0.4) | 1666 | 3932 | 6101 | 10,239 | 24,789 | 46,727 |
| IGA (0.8,0.1,0.1) | 2391 | 3079 | 4455 | 4821 | 5936 | 20,682 |
| IGA (0.7,0.2,0.1) | 1503 | 1926 | 2419 | 3216 | 4234 | 13,298 |
| IGA (0.6,0.2,0.2) | 1100 | 1397 | 1848 | 2193 | 2944 | 9482 |
| IGA (0.5,0.3,0.2) | 840 | 1053 | 1413 | 1652 | 2089 | 7047 |
| IGA (0.4,0.3,0.3) | 653 | 843 | 1192 | 1390 | 1742 | 5820 |
| IGA (0.3,0.3,0.4) | 576 | 699 | 923 | 1123 | 1418 | 4739 |
| PSO-GA (0.6) | 1188 | 1974 | 4049 | 8285 | 7990 | 23,486 |
| GA-PSO (0.3,0.1,0.6) | 894 | 1131 | 2153 | 2357 | 4913 | 11,448 |
| GA-PSO (0.1,0.2,0.7) | 625 | 1212 | 1676 | 2185 | 3463 | 9161 |
| GA-PSO (0.6,0.2,0.2) | 666 | 1236 | 2140 | 2935 | 5261 | 12,238 |
| DPSO (0.2,0.3,0.5) | 432 | 767 | 1136 | 1528 | 2123 | 5986 |
| DPSO (0.3,0.2,0.5) | 495 | 1071 | 1291 | 2070 | 2330 | 7257 |
| DPSO (0.1,0.2,0.7) | 515 | 1047 | 1364 | 1972 | 2398 | 7296 |
| DPSO (0.5,0.1,0.4) | 582 | 1362 | 2089 | 2856 | 3996 | 10,885 |
| DPSO (0.4,0.1,0.5) | 628 | 1354 | 1844 | 2747 | 3974 | 10,547 |
| DPSO (0.3,0.1,0.6) | 540 | 1464 | 1763 | 2623 | 3754 | 10,144 |

### 4.4. Algorithm Experiments

Experiment 1: The objective is to verify the calculation accuracy of the algorithm through comprehensive evaluation value calculations. The parameter groups of all algorithms are calculated according to the iteration number of Formula (25), and the evaluation value is shown in Table 6.

**Table 6.** Multi-algorithm comprehensive evaluation value comparison table.

| Algorithm and Parameter | Number of Equipment Items and Tasks | | | | | |
|---|---|---|---|---|---|---|
| | 8 | 9 | 10 | 11 | 12 | 13 |
| GA (0.6,0.4) | 0.905917 | 0.910875 | 0.901094 | 0.937174 | 0.937185 | 0.936331 |
| IGA (0.8,0.1,0.1) | 0.905917 | 0.910875 | 0.901097 | 0.937207 | 0.937478 | 0.936362 |
| IGA (0.7,0.2,0.1) | 0.905917 | 0.910875 | 0.901097 | 0.937207 | 0.937478 | 0.936362 |
| IGA (0.6,0.2,0.2) | 0.905917 | 0.910875 | 0.901097 | 0.937207 | 0.937478 | 0.936362 |
| IGA (0.5,0.3,0.2) | 0.905917 | 0.910875 | 0.901097 | 0.937207 | 0.937478 | 0.936362 |
| IGA (0.4,0.3,0.3) | 0.905917 | 0.910875 | 0.901097 | 0.937207 | 0.937478 | 0.936362 |
| IGA (0.3,0.3,0.4) | 0.905917 | 0.910875 | 0.901097 | 0.937207 | 0.937478 | 0.936362 |
| PSO-GA (0.6) | 0.905898 | 0.910849 | 0.901045 | 0.936897 | 0.937376 | 0.936174 |
| GA-PSO (0.3,0.1,0.6) | 0.905917 | 0.910875 | 0.901097 | 0.937207 | 0.937458 | 0.936362 |
| GA-PSO (0.1,0.2,0.7) | 0.905917 | 0.910875 | 0.901097 | 0.93715 | 0.937458 | 0.936349 |
| GA-PSO (0.6,0.2,0.2) | 0.905917 | 0.910875 | 0.901097 | 0.937178 | 0.937478 | 0.936349 |
| DPSO (0.2,0.3,0.5) | 0.905917 | 0.910875 | 0.901097 | 0.937207 | 0.937444 | 0.936354 |

**Table 6.** *Cont.*

| Algorithm and Parameter | Number of Equipment Items and Tasks | | | | | |
|---|---|---|---|---|---|---|
| | **8** | **9** | **10** | **11** | **12** | **13** |
| DPSO (0.3,0.2,0.5) | 0.905917 | 0.910875 | 0.901097 | 0.937207 | 0.937386 | 0.936362 |
| DPSO (0.1,0.2,0.7) | 0.905917 | 0.910875 | 0.901097 | 0.937207 | 0.937466 | 0.936362 |
| DPSO (0.5,0.1,0.4) | 0.905917 | 0.910875 | 0.901097 | 0.937207 | 0.937478 | 0.936362 |
| DPSO (0.4,0.1,0.5) | 0.905917 | 0.910875 | 0.901097 | 0.937207 | 0.937478 | 0.936362 |
| DPSO (0.3,0.1,0.6) | 0.905917 | 0.910875 | 0.901097 | 0.937207 | 0.937478 | 0.936362 |

| Algorithm and Parameter | Number of Equipment Items and Tasks | | | | | |
|---|---|---|---|---|---|---|
| | **14** | **15** | **16** | **17** | **18** | **19** |
| GA (0.6,0.4) | 0.920982 | 0.927141 | 0.935498 | 0.937482 | 0.949289 | 0.925388 |
| IGA (0.8,0.1,0.1) | 0.921115 | 0.927263 | 0.936067 | 0.938276 | 0.949914 | 0.925894 |
| IGA (0.7,0.2,0.1) | 0.921115 | 0.927266 | 0.936067 | 0.938287 | 0.949892 | 0.925843 |
| IGA (0.6,0.2,0.2) | 0.921115 | 0.927269 | 0.936067 | 0.938286 | 0.949935 | 0.925872 |
| IGA (0.5,0.3,0.2) | 0.921115 | 0.927269 | 0.936067 | 0.938267 | 0.949858 | 0.92572 |
| IGA (0.4,0.3,0.3) | 0.921115 | 0.927266 | 0.936046 | 0.938245 | 0.949771 | 0.925716 |
| IGA (0.3,0.3,0.4) | 0.921078 | 0.927269 | 0.936063 | 0.938189 | 0.949753 | 0.925521 |
| PSO-GA (0.6) | 0.920734 | 0.92711 | 0.935687 | 0.938034 | 0.949794 | 0.925663 |
| GA-PSO (0.3,0.1,0.6) | 0.921099 | 0.92726 | 0.935907 | 0.938218 | 0.949846 | 0.925759 |
| GA-PSO (0.1,0.2,0.7) | 0.921094 | 0.927193 | 0.935904 | 0.938224 | 0.949888 | 0.925741 |
| GA-PSO (0.6,0.2,0.2) | 0.921027 | 0.927247 | 0.936019 | 0.93824 | 0.949925 | 0.92585 |
| DPSO (0.2,0.3,0.5) | 0.92102 | 0.927258 | 0.936053 | 0.938287 | 0.949983 | 0.925886 |
| DPSO (0.3,0.2,0.5) | 0.921103 | 0.927269 | 0.936067 | 0.938287 | 0.949983 | 0.925906 |
| DPSO (0.1,0.2,0.7) | 0.921115 | 0.927268 | 0.936067 | 0.938287 | 0.949978 | 0.925906 |
| DPSO (0.5,0.1,0.4) | 0.921115 | 0.927269 | 0.936067 | 0.938287 | 0.949983 | 0.925906 |
| DPSO (0.4,0.1,0.5) | 0.921115 | 0.927269 | 0.936067 | 0.938287 | 0.949983 | 0.925906 |
| DPSO (0.3,0.1,0.6) | 0.921115 | 0.927269 | 0.936067 | 0.938287 | 0.949983 | 0.925906 |

| Algorithm and Parameter | Number of Equipment Items and Tasks | | | | | |
|---|---|---|---|---|---|---|
| | **20** | **21** | **22** | **23** | **24** | **25** |
| GA (0.6,0.4) | 0.924698 | 0.93266 | 0.948151 | 0.945635 | 0.937196 | 0.932849 |
| IGA (0.8,0.1,0.1) | 0.942251 | 0.940722 | 0.940895 | 0.948123 | 0.936434 | 0.945138 |
| IGA (0.7,0.2,0.1) | 0.942251 | 0.940707 | 0.940848 | 0.948091 | 0.936418 | 0.945083 |
| IGA (0.6,0.2,0.2) | 0.942251 | 0.940727 | 0.94086 | 0.948109 | 0.936401 | 0.945047 |
| IGA (0.5,0.3,0.2) | 0.942251 | 0.940711 | 0.9408 | 0.948031 | 0.936405 | 0.945035 |
| IGA (0.4,0.3,0.3) | 0.942249 | 0.940695 | 0.940781 | 0.948053 | 0.936265 | 0.944858 |
| IGA (0.3,0.3,0.4) | 0.942251 | 0.940613 | 0.940735 | 0.947955 | 0.936355 | 0.94466 |
| PSO-GA (0.6) | 0.942028 | 0.940462 | 0.940715 | 0.948021 | 0.936212 | 0.944911 |
| GA-PSO (0.3,0.1,0.6) | 0.942157 | 0.940702 | 0.940873 | 0.94802 | 0.936288 | 0.94493 |
| GA-PSO (0.1,0.2,0.7) | 0.942062 | 0.940581 | 0.940739 | 0.947989 | 0.936225 | 0.944912 |
| GA-PSO (0.6,0.2,0.2) | 0.942084 | 0.940555 | 0.94081 | 0.948079 | 0.936324 | 0.945145 |
| DPSO (0.2,0.3,0.5) | 0.94223 | 0.94071 | 0.940962 | 0.9481 | 0.936462 | 0.945255 |

**Table 6.** *Cont.*

| Algorithm and Parameter | Number of Equipment Items and Tasks | | | | | |
|---|---|---|---|---|---|---|
| | **20** | **21** | **22** | **23** | **24** | **25** |
| DPSO (0.3,0.2,0.5) | 0.942249 | 0.940732 | 0.941009 | 0.948124 | 0.936484 | 0.945267 |
| DPSO (0.1,0.2,0.7) | 0.942242 | 0.940732 | 0.941009 | 0.948129 | 0.936482 | 0.945276 |
| DPSO (0.5,0.1,0.4) | 0.942251 | 0.940732 | 0.941009 | 0.948129 | 0.936485 | 0.945276 |
| DPSO (0.4,0.1,0.5) | 0.942251 | 0.940732 | 0.941009 | 0.948129 | 0.936485 | 0.945276 |
| DPSO (0.3,0.1,0.6) | 0.942251 | 0.940732 | 0.941009 | 0.948129 | 0.936485 | 0.945276 |

According to the average value in the above table, a comparison chart of multi-algorithm multi-parameter array can be drawn (Figure 5).

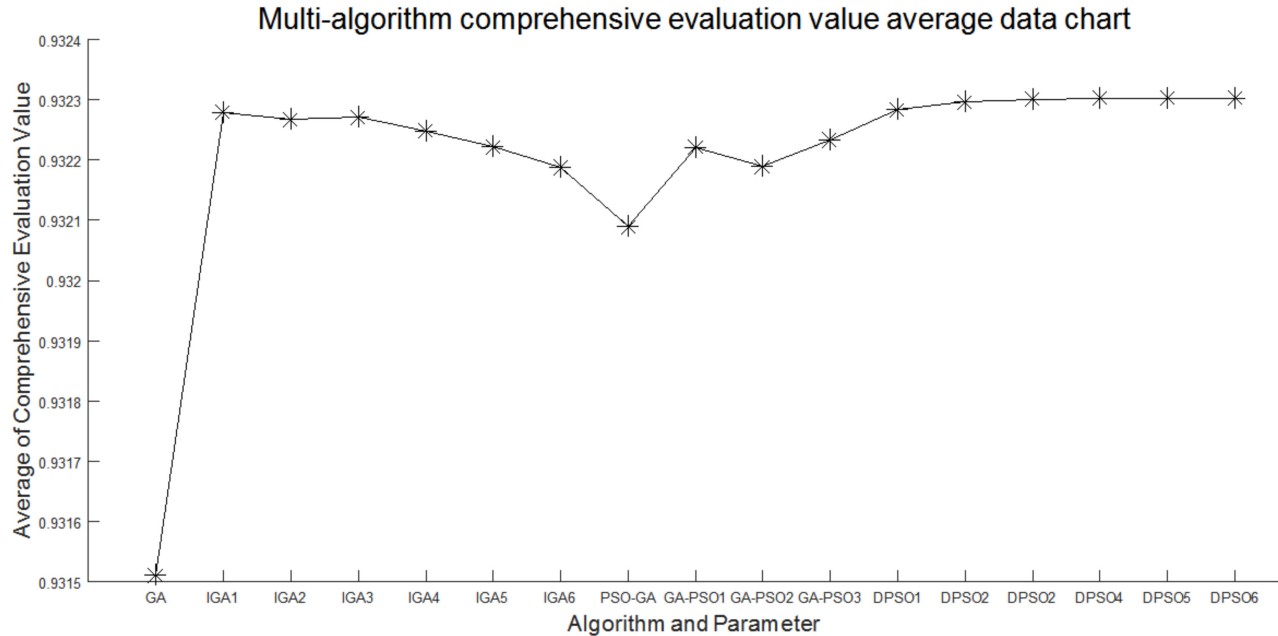

**Figure 5.** Comparison chart of multi-algorithm comprehensive evaluation average.

According to the experiment, it can be seen that the evaluation value obtained by GA is the lowest, followed by PSO-GA, and the best experimental result is DPSO. In all tests of the three parameter groups, the highest value is completely consistent. In this case, it can be considered that the evaluation value calculated according to the three groups of parameters of the algorithm is the global optimal solution.

Experiment 2: the objective is to verify the effectiveness of the algorithm through the success rate. In combination with the results of Experiment 1, three groups of DPSO result data are taken as the global optimal solution and recalculated, and the proportion of the global optimal solution can then be calculated by other algorithms to verify the effectiveness of the algorithm. The algorithm is compared and the best group of algorithm parameters selected from Experiment 1. The obtained experimental data are shown in Table 7.

**Table 7.** Multi-algorithm success rate comparison table.

| Algorithm and Parameter | Number of Equipment Items and Tasks | | | | | | | | |
|---|---|---|---|---|---|---|---|---|---|
| | **8** | **9** | **10** | **11** | **12** | **13** | **14** | **15** | **16** |
| GA (0.6,0.4) | 100% | 100% | 99% | 97% | 89% | 97% | 90% | 81% | 77% |
| IGA (0.8,0.1,0.1) | 100% | 100% | 100% | 100% | 100% | 100% | 100% | 99% | 100% |
| IGA (0.6,0.2,0.2) | 100% | 100% | 100% | 100% | 100% | 100% | 100% | 100% | 100% |
| PSO-GA (0.6) | 99% | 99% | 97% | 95% | 92% | 93% | 88% | 91% | 81% |
| GA-PSO (0.3,0.1,0.6) | 100% | 100% | 100% | 100% | 98% | 100% | 97% | 99% | 94% |
| GA-PSO (0.6,0.2,0.2) | 100% | 100% | 100% | 98% | 100% | 99% | 96% | 96% | 94% |
| DPSO | 100% | 100% | 100% | 100% | 100% | 100% | 100% | 100% | 100% |
| Algorithm and Parameter | Number of Equipment Items and Tasks | | | | | | | | |
| | **17** | **18** | **19** | **20** | **21** | **22** | **23** | **24** | **25** |
| GA (0.6,0.4) | 48% | 46% | 55% | 58% | 21% | 12% | 18% | 10% | 7% |
| IGA (0.8,0.1,0.1) | 97% | 96% | 98% | 100% | 99% | 89% | 98% | 96% | 87% |
| IGA (0.6,0.2,0.2) | 99% | 95% | 96% | 100% | 98% | 86% | 97% | 93% | 77% |
| PSO-GA (0.6) | 84% | 86% | 75% | 84% | 78% | 71% | 82% | 79% | 69% |
| GA-PSO (0.3,0.1,0.6) | 94% | 87% | 84% | 91% | 93% | 84% | 84% | 83% | 72% |
| GA-PSO (0.6,0.2,0.2) | 97% | 91% | 89% | 90% | 86% | 84% | 90% | 86% | 80% |
| DPSO | 100% | 100% | 100% | 100% | 100% | 100% | 100% | 100% | 100% |

According to the above data, a comparison chart can be drawn (Figure 6).

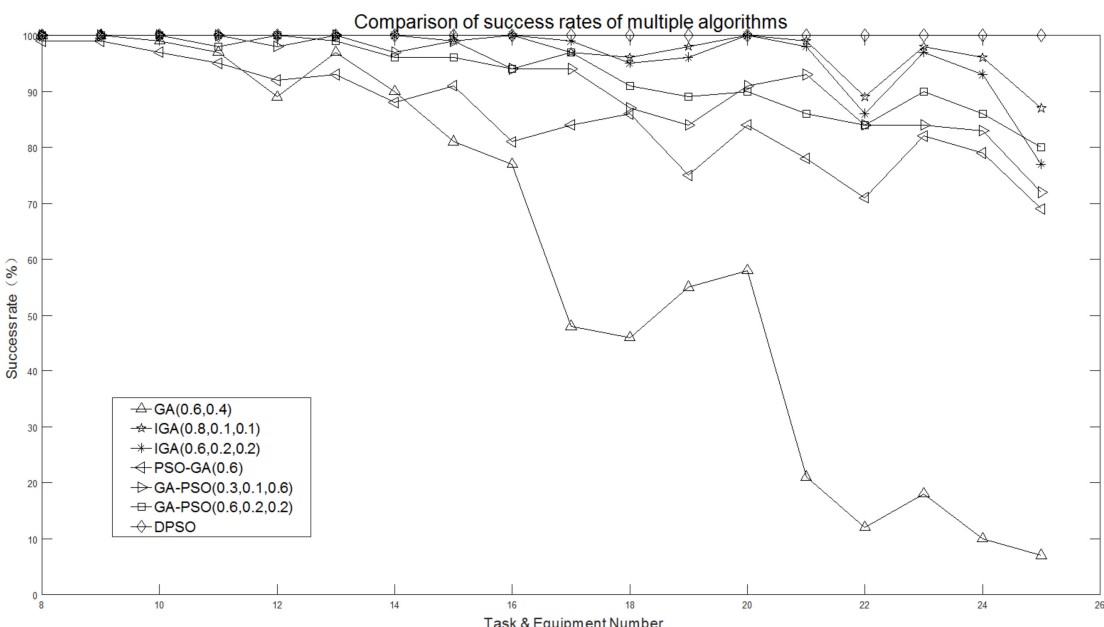

**Figure 6.** Comparison chart of multi-algorithm success rate.

Analysis of experimental data:

1. Except for DPSO, the success rate of all algorithms decreases gradually with the number of tasks.
2. Compared with other algorithms, the GA calculation results are the worst. The IGA, PSO-GA, and GA-PSO calculation results are equivalent, but with the increase in the number of tasks, the gap between the GA and DPSO calculation results is wide.

It shows that the ICO can effectively improve the calculation success rate of the intelligent algorithm.

The number of iterations when the algorithm calculates is recorded to obtain the global optimal solution (if the optimal solution is not obtained, count according to the upper limit of the number of iterations set by the algorithm), and the number of iterations of multiple algorithms is obtained (Table 8).

**Table 8.** Multi-algorithm number of iterations comparison table.

| Algorithm and Parameter | Number of Equipment Items and Tasks | | | | | | | | |
|---|---|---|---|---|---|---|---|---|---|
| | 8 | 9 | 10 | 11 | 12 | 13 | 14 | 15 | 16 |
| GA (0.6,0.4) | 2226 | 4544 | 7299 | 10,334 | 22,411 | 22,703 | 37,218 | 52,963 | 78,680 |
| IGA (0.8,0.1,0.1) | 2152 | 3206 | 3829 | 4906 | 5998 | 6733 | 8202 | 11,879 | 12,107 |
| IGA (0.6,0.2,0.2) | 1016 | 1397 | 1767 | 2390 | 2519 | 3247 | 3910 | 4570 | 5502 |
| PSO-GA (0.6) | 1407 | 2054 | 4339 | 5549 | 11,127 | 11,115 | 20,909 | 21,067 | 41,885 |
| GA-PSO (0.3,0.1,0.6) | 682 | 1282 | 1892 | 2424 | 4489 | 4743 | 7662 | 8182 | 21,488 |
| GA-PSO (0.6,0.2,0.2) | 639 | 984 | 1559 | 3582 | 3776 | 6524 | 9717 | 15,660 | 22,344 |
| DPSO (0.5,0.1,0.4) | 699 | 1226 | 2072 | 2967 | 4438 | 5397 | 7962 | 9579 | 12,810 |
| DPSO (0.4,0.1,0.5) | 622 | 1303 | 1910 | 2879 | 3651 | 4895 | 6265 | 8283 | 10,733 |
| DPSO (0.3,0.1,0.6) | 644 | 1237 | 1874 | 2613 | 3310 | 4407 | 6222 | 7747 | 10,097 |

| Algorithm and Parameter | Number of Equipment Items and Tasks | | | | | | | | |
|---|---|---|---|---|---|---|---|---|---|
| | 17 | 18 | 19 | 20 | 21 | 22 | 23 | 24 | 25 |
| GA (0.6,0.4) | 125,956 | 155,587 | 168,004 | 184,432 | 299,930 | 354,558 | 398,367 | 466,670 | 532,595 |
| IGA (0.8,0.1,0.1) | 20,016 | 26,127 | 29,061 | 18,695 | 29,897 | 73,513 | 46,724 | 59,135 | 125,374 |
| IGA (0.6,0.2,0.2) | 8712 | 19,732 | 22,638 | 9329 | 23,594 | 76,462 | 33,057 | 52,605 | 150,233 |
| PSO-GA (0.6) | 40,362 | 52,858 | 83,780 | 70,156 | 98,626 | 137,713 | 110,321 | 137,800 | 222,637 |
| GA-PSO (0.3,0.1,0.6) | 25,750 | 44,814 | 56,354 | 40,915 | 48,824 | 94,523 | 104,378 | 123,837 | 201,774 |
| GA-PSO (0.6,0.2,0.2) | 19,774 | 43,101 | 51,090 | 56,063 | 73,014 | 114,275 | 81,685 | 116,186 | 176,458 |
| DPSO (0.5,0.1,0.4) | 18,052 | 20,118 | 26,519 | 24,658 | 31,859 | 45,711 | 53,620 | 64,200 | 67,979 |
| DPSO (0.4,0.1,0.5) | 12,995 | 17,680 | 19,686 | 18,324 | 24,852 | 30,707 | 35,037 | 37,732 | 43,879 |
| DPSO (0.3,0.1,0.6) | 13,612 | 14,993 | 17,979 | 17,297 | 21,105 | 27,875 | 30,236 | 32,038 | 45,693 |

According to the above table, the comparison diagram can be drawn (Figure 7). Analysis of the experimental data:

1.  The number of iterations of all algorithms increases gradually with the number of tasks.
2.  The number of iterations of the GA calculation increases rapidly because of the low success rate. In the experiment, it is almost impossible to obtain the optimal solution using the set number of iterations. The number of iterations of PSO-GA and GA-PSO is equal, which shows that both hybrid strategies can effectively improve the convergence of intelligent algorithms. The IGA calculation results are superior to those of the hybrid algorithms. The advantage of the number of iterations of the DPSO calculation is obvious, which proves that the convergence of DPSO is the highest.

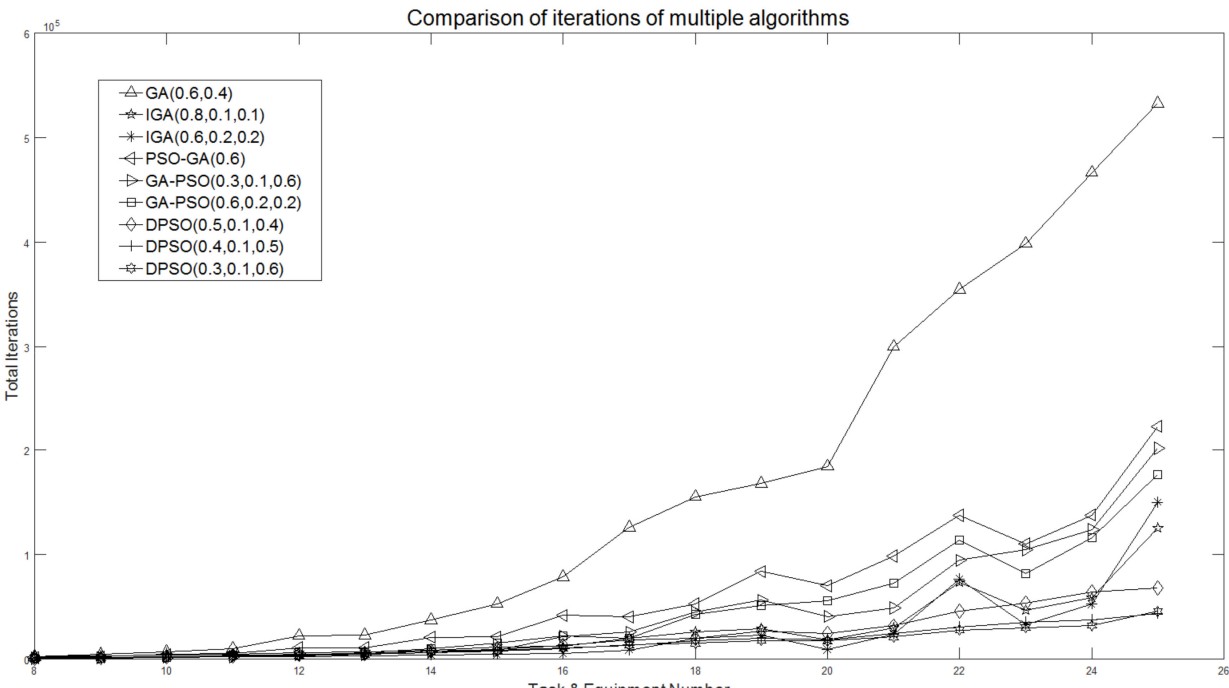

**Figure 7.** Comparison chart of multi-algorithm number of iterations.

*4.5. Discussion of Experiments*

From the experiments, it can be seen that:

1.  The mathematical model and intelligent algorithm built can solve the TTSA of radio and television transmission.

2.  Through the enumeration algorithm verification, when the number of tasks is small, the intelligent algorithm can calculate the global optimal solution, indicating that the intelligent approximation algorithm can achieve the same calculation results as the accurate algorithm. However, the limitation is that not all intelligent algorithms can achieve the same results as accurate algorithms. Comparing different intelligent algorithms is a contribution of this paper.

3.  A large number of attempts of the algorithms can find the optimal approximate calculation scheme for specific problems. The parameter selection of the algorithm is the key to achieving optimal results. This paper directly uses the results of parameter testing in previous studies. It is recommended that when using this algorithm to solve other similar problems, the algorithm parameter selection test ought to be conducted again.

4.  In order to obtain better results, the algorithm should be adapted to specific problems, for example, the improved crossover operator and probability selection model in this paper.

5.  The hybrid algorithm can achieve good results in a certain range, but it may not completely inherit the advantages of the original algorithm. Although the computational results of the PSO-GA and GA-PSO in this paper are better than GA, there is no advantage in comparing the results of IGA and DPSO.

The advantages and disadvantages of the various algorithms and comparison algorithms proposed in this paper are shown in Table 9.

**Table 9.** Comparison of advantages and disadvantages of the algorithms.

| Algorithm | Parameter | Success Rate | Astringency | Accuracy | Comprehensive |
|:---:|:---:|:---:|:---:|:---:|:---:|
| GA | Middle | Low | Low | Low | Low |
| IGA | Middle | Middle | High | High | high |
| PSO-GA | Low | Low | Middle | Middle | Middle |
| GA-PSO | High | Middle | Middle | Middle | Middle |
| DPSO | High | High | Highest | Highest | Highest |

## 5. Conclusions

Based on the analysis of the characteristics of the TTSA, this paper establishes a mathematical model for the quantitative evaluation of the TTSA. Summarizing the previous research results of solving COPs, based on GA and PSO, a variety of intelligent algorithms for solving COPs, such as ICO, hybrid intelligent computing, and redefining the basic operation of PSO, are proposed. Through the analysis and comparison of a large number of simulation experiments, the DSPO that can obtain the global optimal solution within a certain task range was found. The approximate calculation of the algorithm obtained the global optimal solution consistent with the accurate algorithm.

In GA, chromosomes share information with each other, so the movement of the whole population is relatively uniform to the optimal region. In the PSO, only the global optimum and individual optimum give information to other particles, which is a one-way flow of information. The whole search and update process follows the current optimal solution. Compared with GA, in most cases, all particles converge to the optimal solution more quickly. For the TTSA, the individual coding of the population is discrete and has no continuous correlation. The mathematical description of the particle position update when the PSO is used is the key to making full use of the advantages of the PSO. At the same time, the probability model is used to describe the vectorization flight of the particle position, as the proportional and separate position update of the multi-particle population is the key to the success of the DSPO.

Using intelligent algorithms to accomplish TTSA can effectively improve the transmission effect at specific points in time and achieve the goal of optimizing transmission coverage.

The limitation of this paper is that it only considers the static allocation of multiple tasks to multiple pieces of equipment at a single point in time. The next step is to study the dynamic task scheduling based on the TTSA at the same time point, aiming at solving the problem of how to realize the optimal intelligent dynamic task scheduling across time periods when the time span is introduced and the tasks are not transferable. The application scenarios are more in line with the actual situation of current radio and television transmission stations.

**Author Contributions:** Conceptualization, W.Y. and X.W.; methodology, W.Y. and X.W.; software, X.W.; validation, X.W; formal analysis, W.Y.; investigation, W.Y. and X.W.; data curation, X.W.; writing—original draft preparation, X.W.; writing—review and editing, W.Y. All authors have read and agreed to the published version of the manuscript.

**Funding:** This research received no external funding.

**Institutional Review Board Statement:** Not applicable.

**Informed Consent Statement:** Not applicable.

**Data Availability Statement:** Not applicable.

**Conflicts of Interest:** The authors declare no conflict of interest.

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
