# Peer review of "Research on Transmission Task Static Allocation Based on Intelligence Algorithm"

_applsci, doi:10.3390/app13064058_

Round 1

Reviewer 1 Report

The authors developed a hybrid metaheuristic algorithm for the transmission task static assignment problem, which proved to be superior in several numerical cases. The reviewers' comments are as follows.

1. For the interpretation of parameter symbols, there are some formatting problems, where should be lowercase and not indented. In addition, the variables should be italicized.

2. The authors need to proofread and check each formula.

3. Where do the algorithms used for comparison come from?

4. A comparison of the efficiency of the algorithm solutions seems to be missing

5. The conclusion should indicate the limitations of the study and should be more enhanced with a description of the direction of the next in-depth study.

6. There is a problem with the keyword format.

8. As in line 78 ABC, the abbreviations have been expressed in the previous text, and the full name is not needed later.

7. The whole text needs to be proofread and the language checked.

8. The authors should have more detailed summaries of previous studies and explain what gaps their research has filled.

The authors are advised to refer to the following literature to further improve the paper.

[1] Tian, G., Ke, H., & Chen, X. (2014). Fuzzy cost-profit tradeoff model for locating a vehicle inspection station considering regional constraints. Journal of Zhejiang University SCIENCE C, 15(12), 1138-1146.

[2] Tian, G., Zhang, C., Zhang, X., Feng, Y., Yuan, G., Peng, T., and Pham, D. T. (January 19, 2023). "Multi-Objective Evolutionary Algorithm With Machine Learning and Local Search for an Energy-Efficient Disassembly Line Balancing Problem in Remanufacturing." ASME. J. Manuf. Sci. Eng. May 2023; 145(5): 051002. https://doi.org/10.1115/1.4056573

[3] Liu, Q., Liu, Z., Xu, W., Tang, Q., Zhou, Z., & Pham, D. T. (2019). Human-robot collaboration in disassembly for sustainable manufacturing. International Journal of Production Research, 57(12), 4027-4044.

[4] Xu, W., Tang, Q., Liu, J., Liu, Z., Zhou, Z., & Pham, D. T. (2020). Disassembly sequence planning using discrete Bees algorithm for human-robot collaboration in remanufacturing. Robotics and computer-integrated manufacturing, 62, 101860.

Author Response

The authors developed a hybrid metaheuristic algorithm for the transmission task static assignment problem, which proved to be superior in several numerical cases. The reviewers' comments are as follows.

  1. For the interpretation of parameter symbols, there are some formatting problems, where should be lowercase and not indented. In addition, the variables should be italicized.

Modify the format in Tables 1 and Table 2, and modify the variables in italicized for the full text.

  1. The authors need to proofread and check each formula.

Review full-text formulas and modify variable formatting.

  1. Where do the algorithms used for comparison come from?

In 4.2, the source of the algorithm is supplemented, and the reference for the GA are described in Table 3.

  1. A comparison of the efficiency of the algorithm solutions seems to be missing

In 4.3, it is added that the efficiency of the algorithm depends on the calculation of the fitness function. In this paper, the comparison of algorithm iterations is used to replace the comparison of algorithm efficiency.

  1. The conclusion should indicate the limitations of the study and should be more enhanced with a description of the direction of the next in-depth study.

In 5, the limitations of this paper and the content to be studied in the next step are analyzed.

  1. There is a problem with the keyword format.

Adjust keywords and correct formatting errors

  1. As in line 78 ABC, the abbreviations have been expressed in the previous text, and the full name is not needed later.

Full text review, modify the use of abbreviations.

  1. The whole text needs to be proofread and the language checked.

Proofread the full text and correct errors.

  1. The authors should have more detailed summaries of previous studies and explain what gaps their research has filled.

In 5, explain how this article differs from previous studies,explain that improved  genetic algorithms and discrete particle swarm optimization can provide reference for applications in other fields.

The authors are advised to refer to the following literature to further improve the paper.

[1] Tian, G., Ke, H., & Chen, X. (2014). Fuzzy cost-profit tradeoff model for locating a vehicle inspection station considering regional constraints. Journal of Zhejiang University SCIENCE C, 15(12), 1138-1146.

[2] Tian, G., Zhang, C., Zhang, X., Feng, Y., Yuan, G., Peng, T., and Pham, D. T. (January 19, 2023). "Multi-Objective Evolutionary Algorithm With Machine Learning and Local Search for an Energy-Efficient Disassembly Line Balancing Problem in Remanufacturing." ASME. J. Manuf. Sci. Eng. May 2023; 145(5): 051002. https://doi.org/10.1115/1.4056573

[3] Liu, Q., Liu, Z., Xu, W., Tang, Q., Zhou, Z., & Pham, D. T. (2019). Human-robot collaboration in disassembly for sustainable manufacturing. International Journal of Production Research, 57(12), 4027-4044.

[4] Xu, W., Tang, Q., Liu, J., Liu, Z., Zhou, Z., & Pham, D. T. (2020). Disassembly sequence planning using discrete Bees algorithm for human-robot collaboration in remanufacturing. Robotics and computer-integrated manufacturing, 62, 101860.

To study the above literature and cite.

Reviewer 2 Report

Dear authors,

This article presents a discrete version of particle swarm optimization for solving the static task allocation of transmission. The paper is interesting but it has some major review comments that should be addressed as follows:

1- The Abstract has not been written carefully. There are writing and syntax errors such as (line 12). Please improve the writing and the structure of the Abstract. The reviewer is suggesting following the IMRaD abstract style. 

2- In the introduction section. please carefully rewrite the research gaps and contributions, they are confusing due to the way of questions asking rather than the logical "cause-action" relationship.

3- Please write an outline of the rest of the paper as a last paragraph in the introduction section.

4- There are many discrete PSO versions that are available in the literature to solve similar problems, what are the main advantages of your method compared to others? Please discuss! 

5- Success rate and iterations are invalid comparison metrics. As you have all the results, please add the parametric study including the mean, min, max, and standard deviation. 

6- Subsection 4.5 and section 5 are both conclusions, therefore please change the subsection 4.5 title into discussion and improve it by adding major achievements, limitations, and recommendations. 

7- The reference list is limited. The Journal paper should include at least 30 references. 

8- Please further solve some writing problems including grammar, syntax, and typos.  

Best wishes

Author Response

Dear  reviewer,

Thank you for your review, and here are the point-to-point responses:

  • The Abstract has not been written carefully. There are writing and syntax errors such as (line 12). Please improve the writing and the structure of the Abstract. The reviewer is suggesting following the IMRaD abstract style. 

Follow the IMRaD and rewrite the summary as required by the reviewer.

  • In the introduction section. please carefully rewrite the research gaps and contributions, they are confusing due to the way of questions asking rather than the logical "cause-action" relationship.

Revise the introduction section, analyze the main points of the problem, and propose solutions for this article

  • Please write an outline of the rest of the paper as a last paragraph in the introduction section.

Add an outline of other chapters of the paper at the designated location.

  • There are many discrete PSO versions that are available in the literature to solve similar problems, what are the main advantages of your method compared to others? Please discuss! 

Comparing the DPSO algorithm in the literature, this paper uses a probabilistic selection model to cover the motion operations of particle swarm optimization algorithms, increase random perturbation to enhance particle diversity, redefine the location, speed, and target of particles for the transmission task static allocation, and focus on designing a particle update strategy, which is not mentioned in previous literature, and is described in the last paragraph of 1.1.

  • Success rate and iterations are invalid comparison metrics. As you have all the results, please add the parametric study including the mean, min, max, and standard deviation. 

This paper conducts multiple algorithm comparisons, with the goal of obtaining the global optimal solution by calculating each algorithm as the most important comparison indicator. The success rate is the most important comparison indicator, and the algorithm efficiency depends on the calculation speed of the fitness function used in the problem itself. The authors believe that the calculation of the fitness function is the fundamental factor in the efficiency of the algorithm, while the calculation number of fitness functions (the number of iterations of the algorithm) is the most important indicator to measure the efficiency of the algorithm, Due to the large differences between the calculation results of each algorithm, the mean is used for calculation within the algorithm. When selecting the algorithm parameter sets, the maximum success rate and the minimum iteration number are used for measurement. These have been emphasized and added in the paper.

  • Subsection 4.5 and section 5 are both conclusions, therefore please change the subsection 4.5 title into discussion and improve it by adding major achievements, limitations, and recommendations. 

Modify 4.5 and 5.

  • The reference list is limited. The Journal paper should include at least 30 references. 

Added references.

  • Please further solve some writing problems including grammar, syntax, and typos.  

Read and revise the full text carefully, and correct some grammar, syntax, and typos errors.

Best wishes

Round 2

Reviewer 2 Report

The paper can be accepted for publication